# Generation of a Model to Predict Differentiation and Migration of Lymphocyte Subsets under Homeostatic and CNS Autoinflammatory Conditions

**DOI:** 10.3390/ijms21062046

**Published:** 2020-03-17

**Authors:** Catharina C. Gross, Marc Pawlitzki, Andreas Schulte-Mecklenbeck, Leoni Rolfes, Tobias Ruck, Petra Hundehege, Heinz Wiendl, Michael Herty, Sven G. Meuth

**Affiliations:** 1Department of Neurology with Institute of Translational Neurology, University Hospital Münster, Albert-Schweitzer-Campus 1, Building A01, D-48149 Münster, Germany; marc.pawlitzki@ukmuenster.de (M.P.); andreas.schulte-mecklenbeck@ukmuenster.de (A.S.-M.); leoni.rolfes@ukmuenster.de (L.R.); tobias.ruck@ukmuenster.de (T.R.); petra.hundehege@ukmuenster.de (P.H.); heinz.wiendl@ukmuenster.de (H.W.); sven.meuth@ukmuenster.de (S.G.M.); 2Institute of Geometry and Applied Mathematics, RWTH Aachen University, Templergraben 55, D-52056 Aachen, Germany; herty@igpm.rwth-aachen.de

**Keywords:** CNS inflammation, autoimmune diseases, multiple sclerosis, Susac syndrome, cerebrospinal fluid, central nervous system, flow cytometry, Markov chain model

## Abstract

The central nervous system (CNS) is an immune-privileged compartment that is separated from the circulating blood and the peripheral organs by the blood–brain and the blood–cerebrospinal fluid (CSF) barriers. Transmigration of lymphocyte subsets across these barriers and their activation/differentiation within the periphery and intrathecal compartments in health and autoinflammatory CNS disease are complex. Mathematical models are warranted that qualitatively and quantitatively predict the distribution and differentiation stages of lymphocyte subsets in the blood and CSF. Here, we propose a probabilistic mathematical model that (i) correctly reproduces acquired data on location and differentiation states of distinct lymphocyte subsets under homeostatic and neuroinflammatory conditions, (ii) provides a quantitative assessment of differentiation and transmigration rates under these conditions, (iii) correctly predicts the qualitative behavior of immune-modulating therapies, (iv) and enables simulation-based prediction of distribution and differentiation stages of lymphocyte subsets in the case of limited access to biomaterial. Taken together, this model might reduce future measurements in the CSF compartment and allows for the assessment of the effectiveness of different immune-modulating therapies.

## 1. Introduction

The central nervous system (CNS) is an immune-privileged compartment separated from the circulating blood and the peripheral organs by the blood–brain and the blood–cerebrospinal fluid barriers [1]. Under homeostatic conditions these barriers regulate transmigration of distinct immune-cell subsets into the CNS [1]. As a consequence, the immune-cell profiles differ between peripheral blood (PB) and cerebrospinal fluid (CSF) and only distinct immune-cell subsets, mainly fulfilling immune-surveillance functions, are detectable in the CSF of healthy individuals [2,3,4]. Under autoinflammatory conditions, barrier dysfunction and/or hyperactivation of immune-cell subsets results in increased migration of pathogenic lymphocytes into the CNS [5]. Furthermore, autoinflammatory CNS diseases may drive lymphocyte activation/differentiation (e.g., differentiation of CD8^+^ cytotoxic T cells in Rasmussen encephalitis [6] and Susac syndrome (SuS) [7]) and/or intrathecal generation of distinct lymphocyte subsets (e.g., generation of antibody-producing plasmacytoid cells in multiple sclerosis [MS] [8,9,10]). Thus, changes in the composition and activation/differentiation status of lymphocyte subsets in the PB and CSF serve as an indicator of both CNS inflammation and response to immune-modulating therapies. Immune profiling of PB and CSF by multi-parameter flow cytometry provides a deeper insight into the disease underlying pathophysiology [6,7,11,12], thereby supporting differential diagnosis, optimized treatment decisions, and monitoring of treatment-related changes (e.g., treatment responses to natalizumab in MS patients [13,14,15,16]) in autoinflammatory CNS diseases.

Since disease activity and treatment responses are usually monitored using clinical and imaging parameters, repetitive lumbar punctures are difficult to justify. Therefore, CSF samples for longitudinal immune-profiling are often not available, whereas PB is more easily accessible. Thus, a model predicting treatment-related changes in the composition and differentiation state of lymphocytes in the CSF by measurements in the PB is warranted. To overcome the limitation of missing CSF data, mathematical models may allow us to get deeper insights into the kinetics of cell transmigration and distribution within the different compartments [17,18]. In a previous study we have generated a mathematical model for the migration and distribution of CD4^+^ T cell subtypes in the PB and CNS compartment based on the Markov chain theory [17], utilizing experimental data from a widely used blood–brain-barrier model under healthy conditions [19]. Here, our goal is to develop this theoretical model to (i) mirror the relationship of naïve and differentiated lymphocyte subsets in the PB and CSF under homeostatic and autoinflammatory conditions, (ii) predict changes in the peripheral and intrathecal immune-cell profile, and (iii) simulate the intrathecal immune-profile in absence of CSF samples. The obtained quantitative predictions may also be used to compare the effect of different immune-modulating therapies.

## 2. Results

### 2.1. Developing a Model to Predict the Relationship between Naïve and Differentiated Lymphocyte Subsets in the PB and CSF

To develop and validate a model to predict the relationship between naïve and differentiated lymphocyte subsets in the PB and CSF (Figure 1A), 75 patients with somatoform disorders and no signs of an inflammatory CSF (=non-inflammatory controls), 221 therapy naïve relapsing remitting MS (RRMS) patients, 41 RRMS patients under natalizumab, and 28 RRMS patients undergoing alemtuzumab treatment were included in the study (Table 1). Furthermore, PB/CSF pairs and PB of 14 and 19 SuS patients, respectively, were also included (Table 1). The relative proportions of lymphocyte subsets, including naïve HLA-DR^-^ CD4^+^ T-helper cells, cytotoxic HLA-DR^-^ CD8^+^ T cells, CD19^+^ B cells, and CD56^bright^ NK cells, as well as differentiated HLA-DR^+^ CD4^+^ T cells, HLA-DR^+^ CD8^+^ T cells, CD19^low^CD138^+^ plasmacytoid cells, and CD56^dim^ NK cells (Figure 1B) in the PB and CSF were determined by multi-parameter flow cytometry (Appendix A). In accordance with our previous work [19], the behavior of cell differentiation and cell migration is modelled mathematically by a probability distribution of cells at different stages as described in detail in Materials and Methods (Equations (1)–(7)). Here, we consider a total of four possible cell stages, two for the compartments (PB, CSF) and two for naïve and differentiated lymphocytes, and we consider two different transition processes, namely migration and differentiation. We assume linear processes without memory and propose a Markov chain model for the computation of the probability distribution of cells (Figure 1C). The following transition processes are considered: lymphocytes in the PB compartment and naïve stage (stage 1) migrate at transition rate α_1_ to naïve stage cells into the CSF compartment (stage 2). Lymphocytes in the PB compartment and naïve stage (stage 1) transition within this compartment at rate β_1_ into the differentiated stage (stage 3). Lymphocytes in the PB compartment and differentiated stage (stage 3) transition at rate α_2_ into the CSF and stay differentiated (stage 4). Finally, at rate β_2_ naïve lymphocytes transition within the CSF compartment (stage 2) into differentiated ones (stage 4). The transition rate predicts the percentage of cell transition from one stage into another. A higher transition rate indicates that a higher proportion of cells will change their respective stage. The model implicitly assumes a sufficient large number of cells available, so that all transitions between stages can be considered independent of each other.

### 2.2. Model Verification on the Distribution of Lymphocyte Subsets in the PB and CSF under Homeostatic Conditions

First, we validated the model by comparing the predicted cell distributions in PB and CSF compartment under homeostatic conditions with the available data. Multi-parameter flow cytometry of PB and CSF from 75 patients with no sign of an inflammatory CSF as a non-inflammatory control was performed. Flow cytometry of T-lymphocyte subsets in the PB and CSF of these patients revealed that the majority of CD4^+^ and CD8^+^ T cells in the PB were naïve, whereas in comparison the proportions of differentiated HLA-DR^+^ CD4^+^ and particular CD8^+^ T cells were enhanced in the CSF (Figure 2A). This situation was correctly reproduced by the proposed model and reflected by the cell-type specific transition rates (Figure 2B, Table 2). For example, naïve CD8^+^ T cells within the CSF transition with a β_2_ rate of 19.7744 with a 1.9-fold higher probability into differentiated ones than their respective counterparts in the PB (transition rate β_1_ = 10.5938). Moreover, the model predicted the distribution of naïve B-cells in the PB and CSF with high accuracy (Figure 2A). Similar to the in vivo situation, the model showed that proportions of naïve B cells are significantly reduced in the CSF compared to the PB. Although CD19^low^CD138^+^ plasmacytoid cells are absent in the CSF under homeostatic conditions, the model would have predicted occurrence of this subset in the CSF. With regard to NK cells, differentiated CD56^dim^ NK cells were the major NK-cell population in the PB, whereas naïve CD56^bright^ NK cells dominate in the CSF. Again, the model predicted the observed data (status quo) for the PB with a high accuracy and, also, albeit to a lesser extent, in the CSF. Taken together, the proposed model correctly reproduces existing cell population distributions, whereas it is limited with regard to “non-existing” lymphocyte subsets (i.e., plasmacytoid cells); the latter is expected since this scenario is currently not included in the model.

### 2.3. Model Predicting RRMS-Related Changes in Circulating and Intrathecal Lymphocyte Subsets

MS as a paradigmatic example of an autoinflammatory demyelinating CNS disease is characterized by activation of autoreactive T lymphocytes in the periphery resulting in enhanced CNS transmigration [20]. Within the CNS, T lymphocytes are re-activated and cause demyelination and axonal damage [20], resulting in specific disease-related changes, particularly in the intrathecal immune-cell profile (Figure 3A). Overall, frequencies of lymphocytes were increased in the CSF of RRMS patients compared to homeostatic conditions independent of the cell type. Proportions of differentiated CD4^+^ and CD8^+^ T cells were strongly enhanced in the CSF of RRMS patients and to a lower extent in the PB. Furthermore, we detected CD19^low^CD138^+^ differentiated plasmacytoid cells in the CSF of RRMS patients, one of the disease hallmarks [8,9,10]. With regard to NK cells, we confirmed earlier studies by our own group revealing an MS-related decrease of differentiated CD56^dim^ NK cells in the PB (Figure 3A) [21]. MS-related changes were consistently reflected in the simulation results of the proposed mathematical model. Comparison of the calculated transition rates for RRMS patients (Figure 3B, Table 2) with those under homeostatic conditions (Figure 2B, Table 2) revealed that this is due to a change in the migratory capacity of the respective lymphocyte subsets rather than a higher degree of differentiation. Furthermore, the disease-associated change in migratory capacity can also be quantified (Table 2). To test the power of our model, we calculated the lymphocyte distribution in the CSF of five additional therapy naïve RRMS patients, who were not included in our calibration cohort (*N* = 216), based on PB data only, and compared them with the available CSF data (Figure 3C). Although calculated data slightly vary from the actual CSF data on an individual basis, the model overall correctly predicted the distribution of lymphocyte subsets in the CSF compartment.

### 2.4. Model Predictions for Treatment-Related Changes in the Peripheral and Intrathecal Immune Profile

Immune-modulating therapies have distinct effects on the proportions and distributions of lymphocyte subsets in MS (Figure 4A). While natalizumab prevents transmigration of lymphocyte subsets in the CNS by blocking the cell adhesion molecule alpha 4 integrin (Figure 4A, left) [22], alemtuzumab depletes CAMPATH-1 (CD52)-expressing lymphocyte subsets in the periphery (Figure 4A, right) [23]. The proposed model qualitatively and precisely predicts this behavior and we observed decreased proportions of all lymphocyte subsets including plasmacytoid cells in the CSF of natalizumab treated patients (Figure 4B, left), whereas a relative increase of differentiated CD4^+^ and CD8^+^ T cells was observed in the PB and CSF of RRMS patients treated with alemtuzumab (Figure 4 B, right). In contrast, and in accordance with an earlier study by our group [24], alemtuzumab resulted in a relative decrease of differentiated CD56^dim^ NK cells. Changes in the peripheral and intrathecal immune-cell profile were correctly predicted by our model with high accuracy. This example shows that the model also enables a quantitative assessment in addition to the qualitative conclusions by providing factors determining the treatment-dependent changes on migration (α_1,2_) and differentiation (β_1,2_) rates (Figure 4C). A direct comparison of the calculated transition rates revealed that natalizumab reduces the migration rates, whereas alemtuzumab has a higher impact on the differentiation rate (Table 2).

### 2.5. Model Predicting Distribution of Lymphocytes in the CSF Compartment in the Absence of CSF Data

SuS is a CD8^+^-mediated endotheliopathy affecting the microvessels of the brain, eye, and inner ear [7]. Increased proportions of HLA-DR^+^-expressing CD8^+^ T cells in both the PB and CSF is a hallmark of this disease [7]. These changes in the CD8^+^ T-cell compartment compared to its differential diagnosis RRMS are accurately described by our model (Figure 5A) resulting in a higher differentiation rate of particular CD8^+^ T cells in the PB (β_1_ SuS = 22.3179 vs. β_1_ RRMS naïve = 12.6372; Figure 5B, Table 2). The differentiation rate of CD8^+^ T cells in the CSF of SuS patients is only slightly increased (β_2_ SuS = 23.47 vs. β_2_ RRMS naïve = 22.4608) indicating that the increased proportions of differentiated CD8^+^ T cells in the CSF are mainly caused by increased transition of peripheral differentiated CD8^+^ T cells from the PB into the CSF (Figure 5B, Table 2).

Since lumbar puncture is rarely performed in SuS patients for clinical reasons, we used this disease as an example to employ the proposed model for predictions in the case of limited biomaterial. We used PB and CSF flow cytometry data of 14 SuS patients to calibrate the transition rates (Figure 6A, Table 2). Then, we simulated the model to obtain the predicted distribution of naïve and differentiated lymphocyte subsets in the CSF of 19 further SuS with no available CSF data (Figure 6B). Except for plasmacytoid cells, which are absent in the CSF compartment of SuS patients, our model predicted the SuS-related changes in comparison to RRMS patients for all subsets with high accuracy for the 14 SuS patients with available full data sets. Moreover, the model is also able to predict a characteristic immune-cell distribution in the CSF for the 19 SuS patients with available data sets only in the PB.

## 3. Discussion

Despite offering potential treatment targets, the complex processes of transmigration of lymphocyte subsets into the CNS and activation/differentiation within the PB and CSF compartment under homeostatic and autoinflammatory conditions are still poorly understood. We have recently proposed a mathematical framework based on Markov jump processes to the distribution and migration of CD4^+^ T-helper cell subsets in the PB and CNS compartment under homeostatic conditions [19]. Here, we advanced this model to include additional lymphocyte subsets such as cytotoxic CD8^+^ T cells, B cells, and NK cells and extended it to autoinflammatory CNS diseases including MS. We demonstrated that the mathematical model (i) reproduces the acquired PB/CSF data with high accuracy, (ii) provides quantitative factors describing the differentiation and transmigration rate, and (iii) facilitates prediction of data sets that cannot be acquired. In particular, the latter might help to reduce lumbar punctures and flow cytometry processing steps by simulation of the mathematical model.

We developed this model, focusing on the contribution of four processes including (i) transmigration of naïve lymphocytes from the PB across the blood-CSF barrier with the rate α_1_ into the CSF, (ii) differentiation of naïve lymphocytes with the rate β_1_ within the PB, (iii) transmigration of differentiated lymphocytes from the PB with the rate α_2_ into the CSF, and (iv) differentiation of naïve lymphocytes with the rate β_2_ within the CSF compartment. Due to missing markers, recirculation of lymphocytes from the CSF into the PB cannot be incorporated in the model. However, the migration rates α_1_/α_2_ are net rates including both lymphocyte migration into and back from the CSF. Nevertheless, we could demonstrate that our model does not only reproduce the acquired data of the differentiation stage and distribution of T-cell subsets and NK cells in the PB and CSF under steady state conditions, but also in distinct autoinflammatory CNS diseases including MS and SuS with high accuracy. The parameters of the model need to be determined based on existing data sets and, clearly, sufficient training data sets for each disease are a prerequisite. However, this calibration step needs to be conducted only once for each group of patients. Following calibration, the model can be used to also predict previously unknown distributions as exemplified in Figure 6B. Although the model is precise in predicting disease-specific distributions, predicted data may vary from the acquired ones for individual patients, because the model was calibrated using the whole patient cohort. With regard to the B-cell compartment, antibody-producing plasmacytoid cells are only intrathecally generated in MS patients [8,9,10], whereas they are absent under steady state conditions and in SuS [7]. Since our model did not take the de novo generation of distinct lymphocyte subsets into account, it predicted the in vivo situation for plasmacytoid cells with high accuracy for MS patients, and falsely predicted occurrence of these cells in the CSF of non-inflammatory controls and SuS patients. The model may be extended to also include plasmacytoid cells and to improve the prediction in the case of SuS patients at the expense of possibly additional transition processes and additional rates.

An advantage of the explicit mathematical model is that it provides quantitative factors describing the transmigration (α_1_,_2_) and differentiation rates (β_1_,_2_) in addition to the qualitative prognosis as well as changes of these factors by immune-modulating therapies. Moreover, we could demonstrate that the model qualitatively coincides with the expected mechanisms of the immune-modulating therapies. For example, the transmigration blocker natalizumab resulted in a reduction of transmigration rates, whereas alemtuzumab had an impact on the differentiation rates.

Despite being important to predict disease and treatment-related changes in the intrathecal immune-profile, lumbar puncture is not always performed in routine diagnostics, i.e., at first diagnosis of SuS. Using this disease as an example, we could demonstrate that our model facilitates prediction of the distribution and differentiation stages of lymphocyte subsets in the CSF by acquisition of these subsets in the PB. Thus, the model has a high prediction capacity, facilitating reduction in future measurements in certain compartments. For those experiments, the model has been calibrated using a patient cohort (14 SuS patients with full data sets), and it is used for the prediction of individual patients. To eliminate statistical errors, we show a range of expected cell distributions obtained from the prediction of the model for individual patients. Due to low cell numbers and sparse CSF materials, repeated measurements in the CSF are not possible. Thus, we could not generate necessary standard deviations on the supplied patient data. The model could be made robust with respect to variations in the data and the potential impact of the measurement error could also be predicted. Although the specificity of CSF analysis could be enhanced if longitudinal data sets were available [25], for ethical reasons serial lumbar puncture is rarely performed. So far, our model only predicts stationary states, but an extension of this model taking dynamic data sets into account might close this gap.

## 4. Materials and Methods

### 4.1. Ethics Statement

All subjects included in this study gave their written informed consent in accordance with the Declaration of Helsinki and a protocol approved by the Ethics Committee of the University of Münster (registration nos. 2010-262-f-S, approval date: 10th of August 2010; 2012-236-f-S, approval date: 3rd of August 2012;, 2014-068-f-S, approval date: 10th of March 2014; 2016-053-f-S, approval date: 5th of April 2016; and 2019-712-f-S, approval date 18th of December 2019.

### 4.2. Material from Controls and Patients

The blood and CSF samples of control subjects without autoimmune or neuroinflammatory disorders (*N* = 75) were included in this study. All cases presented with non-specific complaints and underwent lumbar puncture during a routine diagnostic examination conducted to rule out any neurological condition. None of the healthy controls suffered from a neurological disorder, nor did they show any specific abnormalities during the neurological examination. In addition to the clinical classification, controls fulfilled the following laboratory criteria defining a non-inflammatory CSF: <5 cells/μL CSF, <20 mM lactate in the CSF, no dysfunction of the blood/CSF barrier (defined by the age-adjusted albumin CSF/serum quotient), no oligoclonal bands (OCBs) in the CSF, and no intrathecal immunoglobulin (Ig)G, IgA, or IgM synthesis.

Moreover, we included *N* = 290 RRMS patients according to 2017 revised McDonald criteria [26], from which *N* = 221 were treatment naïve, *N* = 41 were under Natalizumab treatment and *N* = 28 were under Alemtuzumab therapy. In addition, *N* = 33 SuS patients, according to the recently defined criteria, were included [27]. For more details, see Table 1.

### 4.3. Flow Cytometry

Single-cell suspensions from human peripheral blood mononuclear cells and CSF cells were stained for 30 min at 4 °C with the appropriate combination of indicated fluorescence-labeled monoclonal antibodies in PBS, containing 0.1% sodium azide and 0.1% bovine serum albumin (BSA) following treatment with VersaLyse (Beckman Coulter GmbH, Krefeld, Germany) according to the manufacturer’s instructions. The following monoclonal antibodies were used at 1:200 dilutions: CD14-FITC, CD138-PE, HLA-DR-ECD, CD3-PC5.5, CD56-PC7, CD4-APC, CD19-APCAlexafluor700, CD16-APCAlexafluor750, CD8-PacificBlue, and CD45-KromeOrange (all Beckman Coulter). Flow cytometric analysis of stained cells was performed following standard protocols. Cells were analyzed on a Navios™ flow cytometer (Beckman Coulter) using Kaluza Analysis Software (V2.1, Beckman Coulter) and presented using Prism 6.0 (Graph Pad). 

### 4.4. Gating Strategy

Leukocytes from the PB and CSF were gated based on established lineage markers for major cell populations (Appendix A). Therefore, leukocytes were identified based on forward-scatter channel and CD45-expression characteristics. Lymphocytes were selected from total leukocytes as side-scatter channel (SSC)^low^ CD14^-^ cells. Lymphocytes were further split into T cells (CD3^+^CD56^-^), NK cells (CD3^-^CD56^+^), B cells (CD19^+^CD138^-^ naïve B cells), and plasmacytoid cells (CD19^low^CD138^+^, differentiated B cells). In addition, NK-cell subsets were defined by differential expression of CD56 and CD16 as CD56^bright^CD16^dim/-^ (naïve) and CD56^dim^CD16^+^ (differentiated) NK cells. Finally, CD4^+^CD8^-^ T-helper and CD4^-^CD8^+^ cytotoxic T-cell subsets were selected from total T cells and split into HLA-DR^-^ naïve as well as HLA-DR^+^ differentiated cells, respectively. For quantification of total cell counts, FlowCount Fluorospheres (Beckman Coulter) were selected based on fluorescence characteristics.

### 4.5. Mathematical Modelling

We propose a cell transition model describing the probability of naïve lymphocytes to transmigrate into the CSF or differentiate. The model can be rigorously derived using Markov jump processes describing differentiation and migration. A detailed derivation has been shown in Ruck et al. [19]. Therein, the model was derived by first principles and by applying techniques developed for large particle systems in gas dynamics [28]. We propose a novel model in a similar spirit. A formal derivation can be done analogously to the procedure outlined in [19]. We assume four possible stages, two for the compartments (PB, CSF) and two for naïve and differentiated lymphocytes. In each stage lymphocytes undergo a transition to another stage within a certain time interval with a certain probability. For example, a naïve lymphocyte from the PB migrates into the CSF while remaining naïve with probability p = α_1_ (Figure 1A,C). We model the transition of lymphocytes to different stages as a time-independent process. Furthermore, we assume that the cells transition independently of each other, because there are sufficient cells available. The transition model contains jump (i.e., transition) rates that are calibrated using experimental data as described below.

In more detail, we proceed as follows: The quantity we model are cell distributions. We assume that cells at different stages undergo different processes. A total of four possible cell stages X in R^4^, two for the compartments (PB, CSF) and two for naïve and differentiated lymphocytes, respectively, are considered. Transition between all stages will be described next. We assume linear, independent transition processes without memory. Hence, the model for transition between stages is the Markov chain model and the modelled transitions are shown in Figure 1C. Lymphocytes in the compartment PB and naïve stage transition with the rate α_1_ to naïve stages in the compartment CSF. Lymphocytes in the compartment PB and stage naïve transition within this compartment with rate β_1_ into the differentiated stage. Lymphocytes in the compartment PB and differentiated stage transition with rate α_2_ into the CSF (and stay differentiated). Finally, with the rate β_2_ naïve lymphocytes transition within the CSF compartment into differentiated ones. We assume a sufficiently large number of cells available such that transition between stages can be considered independent, and we use the steady state of Markov process for prediction of the cell distribution.

The model can be stated in the mathematical equations as follows: Denote by X in R^4^ the probability to find a cell in one of the stages (X_1_ = PB/naïve, X_2_ = PB/differentiated, X_3_ = CSF/naïve, X_4_ = CSF/differentiated). Then, the number of cells X_i_ A0 in each stage i = 1,..,4 is obtained upon multiplication with the total number of cells A0. Since X_1_,.., X_4_ are probabilities we have X_1_ + X_2_ + X_3_ + X_4_ = 1. Furthermore, X_1_,.., X_4_, fulfills the following equations:X_1_ = (1 − α_1_ − β_1_) A0(1)
(1 + β_1_) X_2_ = α_1_ A0(2)
(1 + α_2_) X_3_ = β_1_ A0(3)
X_4_ = β_1_ X_2_ + α_2_ X_3_(4)
The equation can be rewritten in compact form as
A(α, β) X = b(α, β)(5)

For some invertible 4 × 4 matrix A and a 4 × 1 vector b, both depending linearly on α_1_, α_2_, β_1_, and β_2_. Provided we have k = 1,..,K, experimental measurements of cell distributions in all four stages given, i.e., X_k_ in R^4^, we use the model to estimate the rates α_1_, α_2_, β_1_, and β_2_. Since we do not have information on possible measurement error on the experimental data we consider an unweighted and unbiased least-square formulation to obtain optimal rates α = (α_1_, α_2_) and β = (β_1_, β_2_). To this end we solve the following problem:(6)$$\frac{1}{K}\overset{K}{\underset{K=1}{\sum }}||X\left(\alpha ,\beta \right)-X_{K}|{|}^{2}\to \begin{array}{c} min\\ \alpha ,\beta \end{array}$$
where
(7)$$X\left(\alpha ,\beta \right)=A^{-1}\left(\alpha ,\beta \right)b\left(\alpha ,\beta \right)$$
and where we additionally impose the constraints 0 ≤α,β≤1.

Even if A, b are linearly dependent on α_1_, α_2_, β_1_, and β_2_, the previous problem is nonlinear due to the occurrence of the inverse of the matrix A(α, β). It is easy to show that A(α, β) is invertible for all 0 < α_1_, α_2_, β_1_, β_2_ <1. The minimization problem is solved numerically up to a tolerance of 10^−16^. The problem did have a numerical solution for each data set we tested.

## Figures and Tables

**Figure 1 ijms-21-02046-f001:**
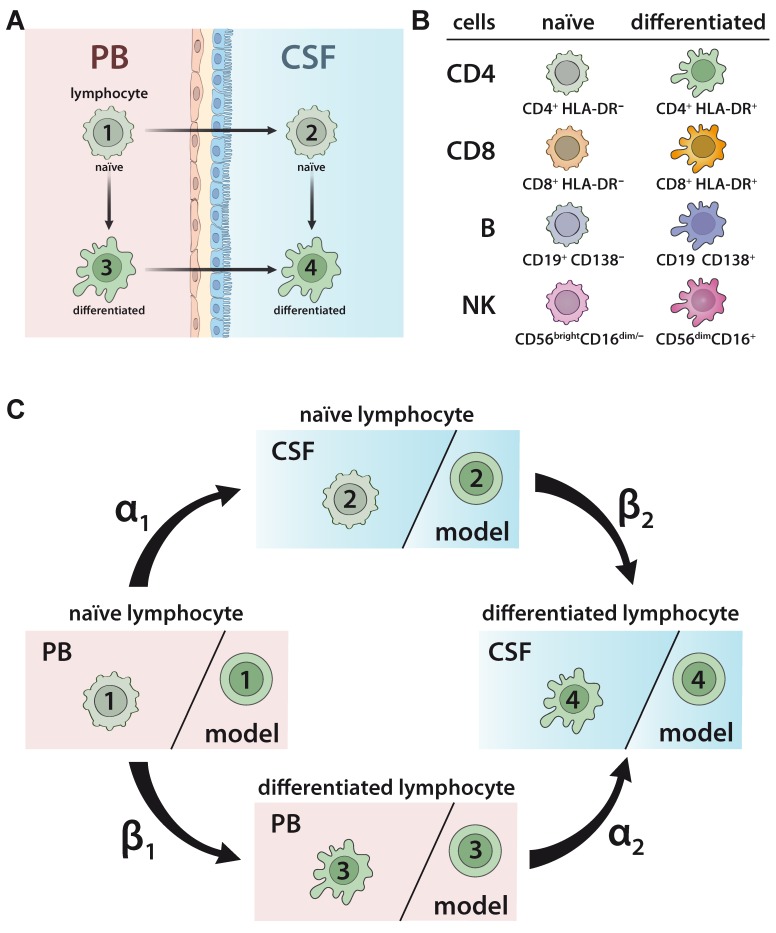
Model describing the relationship between naïve and differentiated lymphocyte subsets in the peripheral blood (PB) and cerebrospinal fluid (CSF). (**A**) Scheme illustrating the relationship between naïve and differentiated lymphocytes in the peripheral blood (PB) and cerebrospinal fluid (CSF). Naïve lymphocytes (stage 1) can either migrate from the PB compartment into the CSF compartment (transition from stage 1 to stage 2) or differentiate within the PB compartment (transition from stage 1 to stage 3). Differentiated lymphocytes (stage 3) can migrate from the PB compartment into the CSF compartment (transition from stage 3 to stage 4). Finally, within the CSF compartment, naïve lymphocytes (stage 2) can differentiate (transition from stage 2 to stage 4). (**B**) Scheme of lymphocyte subsets investigated in the study. The proportions of the following cell populations were investigated in the PB and CSF: CD4^+^HLA-DR^-^ and CD8^+^HLA-DR^-^ naïve T-helper and cytotoxic T-cell subsets, respectively; CD4^+^HLA-DR^+^ and CD8^+^HLA-DR^+^ differentiated T-helper and cytotoxic T-cell subsets, respectively; CD19^+^CD138^-^ naïve B cells and CD19^low^CD138^+^ differentiated plasmacytoid cells; and CD56^bright^CD16^dim/-^ naïve and CD56^dim^CD56^bright^ differentiated NK cells. (**C**) Illustration of Markov chain model for prediction of the relative distribution of naïve and differentiated lymphocyte subsets in PB and CSF. The rates indicate the transitions (%) from (i) naïve lymphocytes in the PB compartment to the CSF compartment (α_1_), (ii) naïve lymphocytes to differentiated lymphocytes within the PB (β_1_), (iii) differentiated lymphocytes from the PB to the CSF compartment (α_2_), and (iv) naïve lymphocytes to differentiated lymphocytes within the CSF (β_2_).

**Figure 2 ijms-21-02046-f002:**
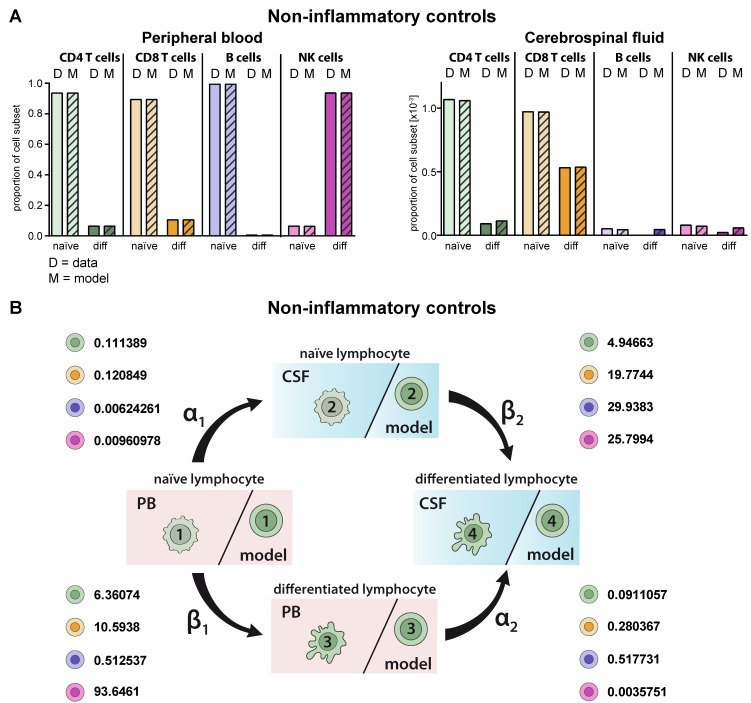
Generation of a model describing the relative distribution of naïve and differentiated lymphocyte subsets in the PB and CSF of non-inflammatory controls. (**A**) Graphs showing the relative abundance of naïve (light colors) and differentiated (diff, dark colors) lymphocyte subsets (D = data) including CD4^+^ T cells (green), CD8^+^ T cells (orange), B cells (violet), and NK cells (purple) in the PB (top) and CSF (bottom) of non-inflammatory controls (*N* = 75). Mean values ± SD of data are provided in Appendix A. Data were used to derive a model (M = model, hatched bars) for lymphocyte-subset specific differentiation and transmigration into the CSF. (**B**) Rates (%) determined by the model indicating the transitions from (i) naïve lymphocytes in the PB compartment to the CSF compartment (α_1_), (ii) naïve lymphocytes to differentiated lymphocytes within the PB (β_1_), (iii) differentiated lymphocytes from the PB to the CSF compartment (α_2_), and (iv) naïve lymphocytes to differentiated lymphocytes within the CSF (β_2_) under steady state conditions are displayed for the respective lymphocyte subsets.

**Figure 3 ijms-21-02046-f003:**
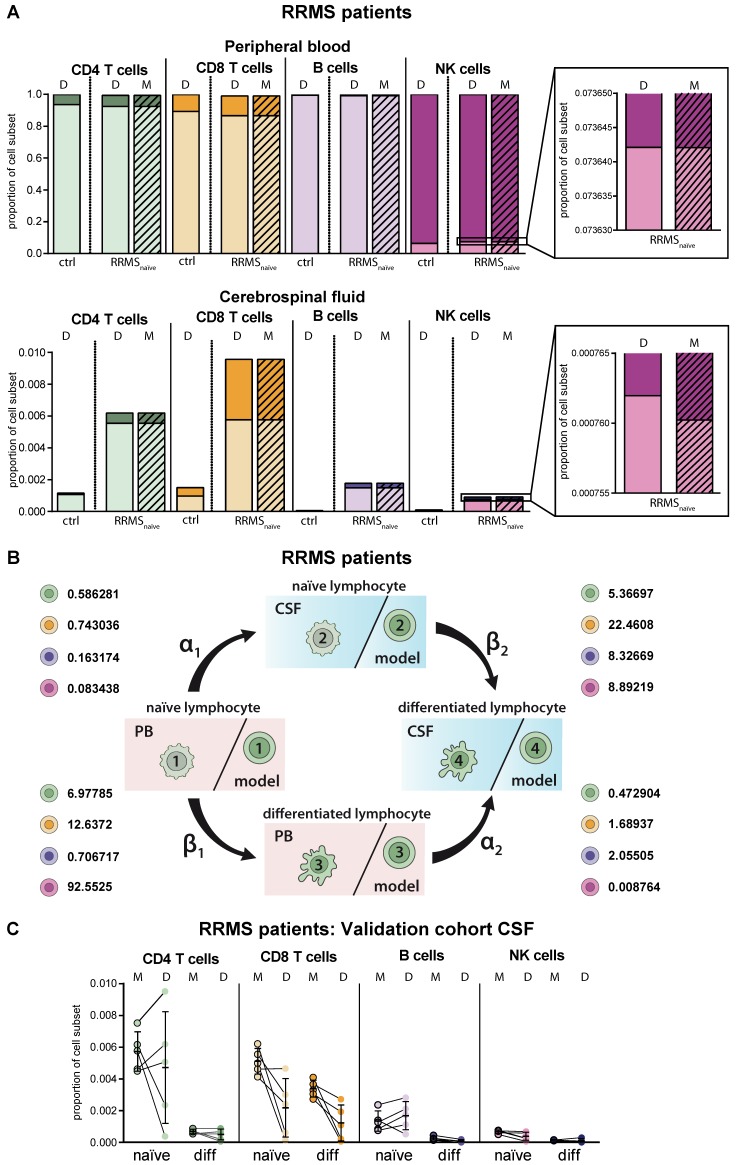
Adaptation of the model for predicting lymphocyte subset differentiation and transmigration in CNS autoimmunity. (**A**) Comparison of naïve (light colors) and differentiated (dark colors) lymphocyte-subset derived data (D, Appendix A) from treatment-naïve RRMS patients (*N* = 216) and data calculated by the model (M, hatched bars) with data derived from non-inflammatory controls (*N* = 75) in the PB (top) and CSF (bottom). Boxes show an enhanced section of the indicated NK-cell bars. (**B**) Rates (%) determined by the model indicating the transitions from (i) naïve lymphocytes in the PB compartment to the CSF compartment (α_1_), (ii) naïve lymphocytes to differentiated lymphocytes within the PB (β_1_), (iii) differentiated lymphocytes from the PB to the CSF compartment (α_2_), and (iv) naïve lymphocytes to differentiated lymphocytes within the CSF (β_2_) of RRMS patients are displayed for the respective lymphocyte subsets. (**C**) Example of additional RRMS patients (*N* = 5), who were not included in calibration of the model and the distribution of each lymphocyte subset was calculated based on the PB data only. CSF data calculated by the model (M) and the respective CSF data (D) are displayed for each patient. Mean values and SD are displayed.

**Figure 4 ijms-21-02046-f004:**
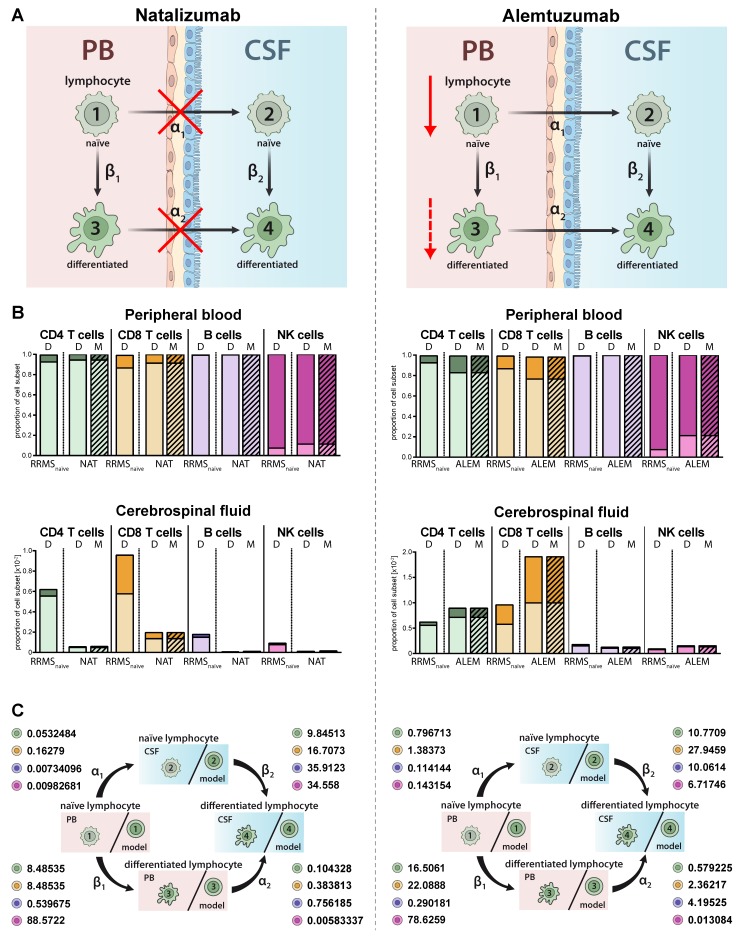
Adjustment of the model to describe alterations in lymphocyte subset transmigration and differentiation in RRMS patients. (**A**) Illustration of the mode of action of the transmigration-blocker natalizumab (left) and lymphocyte-depleting agent alemtuzumab (right). (**B**) Graphs displaying the effect of immune-modulating treatment on RRMS patients (*N* = 216) with natalizumab (left, NAT, *N* = 41) and alemtuzumab (right, ALEM, *N* = 28). Data (D, Appendix A) showing the frequency of naïve (light colors) and differentiated (dark colors) lymphocyte subsets in the PB (top) and CSF (bottom) are compared to values calculated by the model (M, hatched bars). (**C**) Rates (%) determined by the model indicating the transitions from (i) naïve lymphocytes in the PB compartment to the CSF compartment (α_1_), (ii) naïve lymphocytes to differentiated lymphocytes within the PB (β_1_), (iii) differentiated lymphocytes from the PB to the CSF compartment (α_2_), and (iv) naïve lymphocytes to differentiated lymphocytes within the CSF (β_2_) in RRMS patients under natalizumab (left) or alemtuzumab (right) treatment are displayed for the respective lymphocyte subsets.

**Figure 5 ijms-21-02046-f005:**
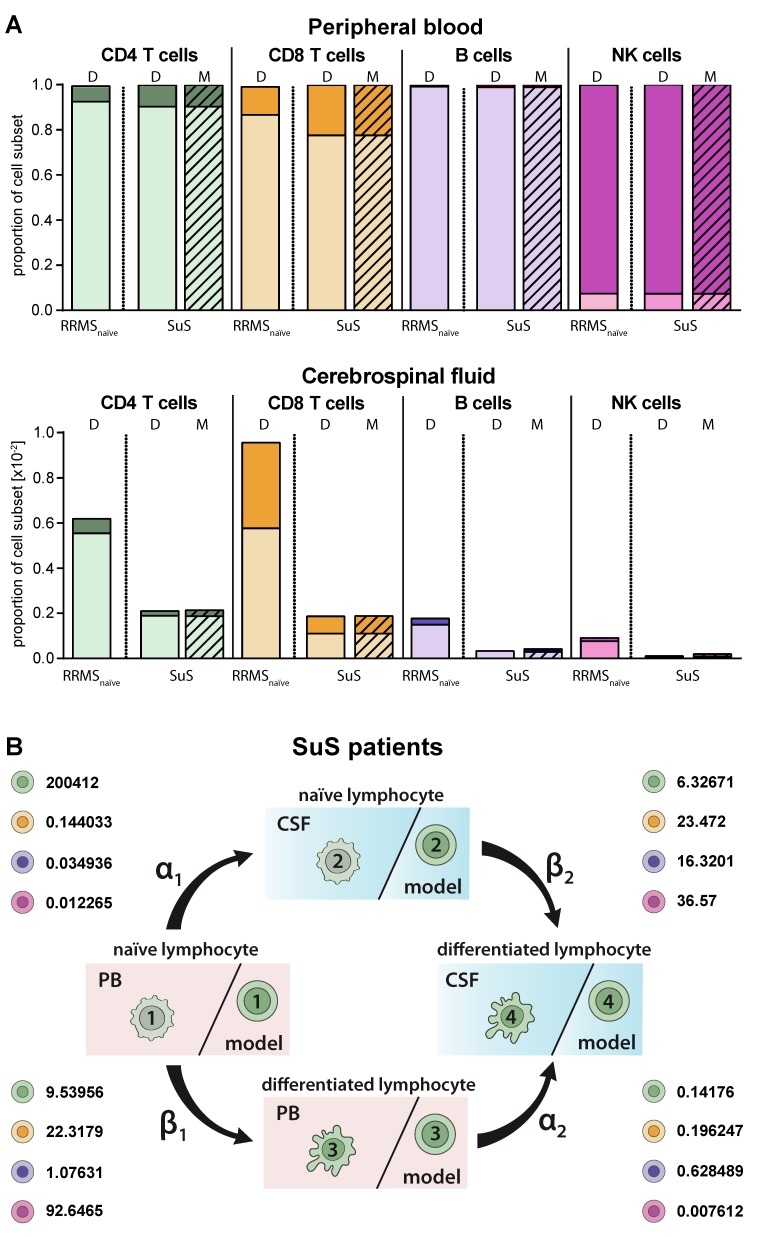
Adjustment of the model to describe alterations in lymphocyte subset transmigration and differentiation in SuS patients. (**A**) Comparison of naïve (light colors) and differentiated (dark colors) lymphocyte-subset derived data (D, Appendix A) from SuS patients (*N* = 14) and data calculated by the model (M, hatched bars) with data derived from therapy naïve RRMS patients (*N* = 216) in the PB (top) and CSF (bottom). (**B**) Rates (%) determined by the model indicating the transitions from (i) naïve lymphocytes in the PB compartment to the CSF compartment (α_1_), (ii) naïve lymphocytes to differentiated lymphocytes within the PB (β_1_), (iii) differentiated lymphocytes from the PB to the CSF compartment (α_2_), and (iv) naïve lymphocytes to differentiated lymphocytes within the CSF (β_2_) of SuS patients are displayed for the respective lymphocyte subsets.

**Figure 6 ijms-21-02046-f006:**
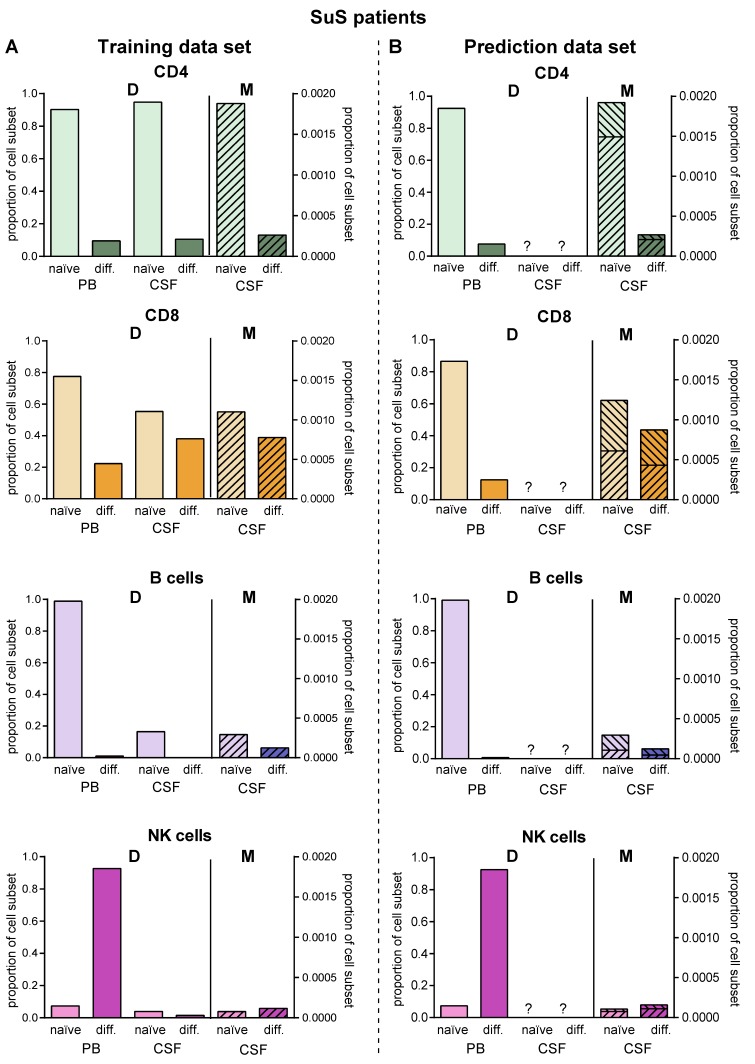
Validation of the model. (**A**) Comparison of data (D, Appendix A) showing frequencies of naïve (light colors) and differentiated (dark colors) lymphocyte subsets from patients with SuS (*N* = 14) in PB and CSF with data calculated by the model (M, hatched bars) in the CSF. (**B**) The derived model was used to predict proportions (M, hatched bars) of naïve (light colors) and differentiated (dark colors) lymphocyte subsets in the CSF of SuS patients based on available data (D, Appendix A) from naïve (light colors) and differentiated (dark colors) lymphocyte subsets in PB of SuS patients (*N* = 19) with no information on the distribution of naïve (light colors) and differentiated (dark colors) lymphocyte subsets in the CSF. The ranges indicate the variance resulting from the fact that the model was generated using the whole training cohort, where it was then used to calculate CSF cell numbers in the prediction cohort.

**Table 1 ijms-21-02046-t001:** Clinical characteristics of the examined groups.

Group	*N*	Females (%)	Age (Median/IQR)
Controls	75	62.7%	33.9/21.4
RRMS naïve	221	74.0%	31.9/17.2
RRMS NAT	41	73.2%	35.1/17.4
RRMS ALEM	28	60.7%	35.8/15.9
SuS	33	72.7%	34.7/13.7

*N* = number, ALEM = alemtuzumab, IQR = interquartile range, NAT = natalizumab, RRMS = relapsing remitting multiple sclerosis, SuS = Susac syndrome.

**Table 2 ijms-21-02046-t002:** Transition rates of lymphocyte subsets based on Markov chain model. The rates indicate the percentage of cells transitioning from (i) naïve lymphocytes in the PB compartment to the CSF compartment (α_1_), (ii) naïve lymphocytes to differentiated lymphocytes within the PB (β_1_), (iii) differentiated lymphocytes from the PB to the CSF compartment (α_2_), and (iv) naïve lymphocytes to differentiated lymphocytes within the CSF (β_2_).

		Transition Rates (%)				
Cells		Controls	RRMS Naïve	RRMS NAT	RRMS ALEM	SuS
**CD4**	α1	0.111389	0.586281	0.0532484	0.796713	0.200412
	α2	0.0911057	0.472904	0.104328	0.579225	0.14176
	β1	6.36074	6.97785	5.56686	16.5061	9.53956
	β2	4.94663	5.36697	9.84513	10.7709	6.32671
**CD8**	α1	0.120849	0.743036	0.16279	1.38373	0.144033
	α2	0.280367	1.68937	0.383813	2.36217	0.196247
	β1	10.5938	12.6372	8.48535	22.0888	22.3179
	β2	19.7744	22.4608	16.7073	27.9459	23.472
**B**	α1	0.00624261	0.163174	0.00734096	0.114144	0.034936
	α2	0.517731	2.05505	0.756185	4.19525	0.628489
	β1	0.512537	0.706717	0.539675	0.290181	1.07631
	β2	29.9383	8.32669	35.9123	10.0614	16.3201
**NK**	α1	0.00960978	0.083438	0.00982681	0.143154	0.012265
	α2	0.0035751	0.008764	0.00583337	0.013084	0.007612
	β1	93.6461	92.5525	88.5722	78.6259	92.6465
	β2	25.7994	8.89219	34.558	6.71746	36.57

ALEM = alemtuzumab, NAT = natalizumab, RRMS = relapsing remitting multiple sclerosis, SuS = Susac syndrome.

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
