# Peer review of "Generation of a Model to Predict Differentiation and Migration of Lymphocyte Subsets under Homeostatic and CNS Autoinflammatory Conditions"

_ijms, 2020, doi:10.3390/ijms21062046_

Round 1
Reviewer 1 Report
The manuscript “generation of a model to predict differentiation and migration of lymphocyte subsets under homeostatic and CNS autoinflammatory conditions” by Gross et al. offers an interesting approach to mathematically model relations of peripheral blood and CSF immune-cell subsets. This is of particular importance because it offers the possibility to draw conclusions on CSF cell numbers without the necessity of repeated CSF draws, which are clinically not feasible. The authors use a large number of matched peripheral blood and CSF from healthy/ non-neurological subjects in comparison to treated and untreated MS patients and Susac syndrome patients to validate their model.
Overall, this is a very interesting paper and it is reassuring that the model shows precise predictions with few exceptions.
The main questions are:
- Why is the transition from the CSF to the PB not included?
- What exactly does the transition rate describe? Is it the percentage of cells that transition to the (e.g.) differentiated cell type? However, this doesn’t fit with the CSF values in Figure 1. Please explain.
- Patient based data should be presented with error bars or as dot blots.
The paper requires edits of data presentation and description to make it accessible to the readers.
- The manuscript requires proofreading and correction of grammatical errors, punctuation and typos.
- The first paragraph of the mathematical modeling section (methods) is jargon heavy and does not sufficiently convey a message to the general reader population.
- The sentence regarding CD19low cells (l. 203) does not make any sense and does not match the figure. Are cells possibly absent and not abundant?
- Spell out abbreviations when a term is first used (RRMS).
- Supplementary figure: To clarify gating strategy, add arrows to make succession more clear.
- Line 260: Do the authors mean CD56 instead of CD52?
- Figure 4: It would be better to stick to a schematic alpha/ beta instead of introducing ABCD
What exactly does the transition rate reflect? Percentages? Frequencies? High numbers indicate increase ? This needs to be explained.
Transition from naivle to diff in the peripheral blood: CD8 cells have a rate of 10, less diff than naïve. NK cells have a rate of 93, more diff tthan naïve. Numbers need a context.
Author Response
Reviewer 1:
The manuscript “generation of a model to predict differentiation and migration of lymphocyte subsets under homeostatic and CNS autoinflammatory conditions” by Gross et al. offers aninteresting approach to mathematically model relations of peripheral blood and CSF immunecell subsets. This is of particular importance because it offers the possibility to draw conclusions on CSF cell numbers without the necessity of repeated CSF draws, which are clinically not feasible. The authors use a large number of matched peripheral blood and CSF from healthy/ non-neurological subjects in comparison to treated and untreated MS patients
and Susac syndrome patients to validate their model.
Overall, this is a very interesting paper and it is reassuring that the model shows precise predictions with few exceptions.
We would like to thank the reviewer for his/her helpful comments.
The main questions are:
(1) Why is the transition from the CSF to the PB not included?
Due to missing markers we cannot determine the number of cells that transition back from the CSF into the PB. However, the calculated a1/a2 migration rates are net rates that are a combination of cells transition into the CSF and from the CSF to the PB. To clarify this point we changed the respective part in the discussion section as followed: “Due to missing markers, recirculation of lymphocytes from the CSF into the PB cannot be incorporated in the model.
However, the migration rates a1/a2 are net rates including both lymphocytes migration into and back from the CSF.
(2) What exactly does the transition rate describe? Is it the percentage of cells that transition to the (e.g.) differentiated cell type? However, this doesn’t fit with the CSF values in Figure 1. Please explain.
What exactly does the transition rate reflect? Percentages? Frequencies? High numbers indicate increase? This needs to be explained. Transition from naïve to diff in the peripheral blood: CD8 cells have a rate of 10, less diff than naïve. NK cells have a rate of 93, more diff than naïve. Numbers need a context.
The transition rate describes the percentage of cells transition from one stage into another. A higher transition rate indicates that a higher proportion of cells will change their respective stage. This fits well with the CSF values in Fig. 2 (since no values are shown in Fig. 1 we assume that the reviewer is referring to Fig. 2). For instance, CD8+ T cells are more differentiated in the CSF compared to the PB, which is reflected by a higher transition rate of this cell population (ß2 = 19.7449) in the CSF than in the PB (ß1 = 10.5938). On the other hand, the proportion of naïve CD56bright NK cells is increased in the CSF compartment compared to PB, which is reflected by a lower transition rate (ß2 = 25.7994) in the CSF than in the PB (ß1 = 93.6461). Moreover, to stick to the example provided by the reviewer: naïve NK cells in the peripheral blood will transition 9.3 x more into differentiated ones than naïve CD8+ T cells. We have now changed the result part accordingly to clarify this point.
To clarify this point we have now added the following sentence: “Thus, the transition rate describes the percentage of cells transition from one stage into another. A higher transition rate indicates that a higher proportion of cells will change their respective stage.”
Furthermore, we included the following examples to bring the rates into a better context: “For example, naïve CD8+ T cells within the CSF transition with a ß2 rate of 19.7744 with a 1.9 x higher probability into differentiated ones than their respective counterparts in the PB (transition rate ß1 = 10.5938).”
“Increased proportions of HLA-DR+ expressing CD8+ T cells in both the PB and CSF is a hallmark of this disease [7]. These changes in the CD8+ T-cell compartment compared to its differential diagnosis RRMS are accurately described by our model (Fig. 5A) resulting in a higher differentiation rate of particular CD8+ T cells in the PB (b1 SuS = 22.3179 vs. b1 RRMS naïve = 12.6372) (Fig. 5B, Table 2). The differentiation rate of CD8+ T cells in the CSF of SuS patients is only slightly increased (b2 SuS = 23.47 vs. b2 RRMS naïve = 22.4608) indicating that the increased proportions of differentiated CD8+ T cells in the CSF are mainly caused by increased transition of peripheral differentiated CD8+ T cells from the PB into the CSF (Fig. 5B, Table 2).”
(3) Patient based data should be presented with error bars or as dot blots.
We agree with the reviewer that this important information should be included in the manuscript. However, since we used stacked bars, figures are difficult to read when error bars are included within the figure. Therefore, we provided an additional supplemental table 1 showing the mean values ± SD of the patient-based data.
(4) The paper requires edits of data presentation and description to make it accessible to the readers.
We apologize for the quality of the figures in the last version and have now uploaded new highquality figures.
(5) The manuscript requires proofreading and correction of grammatical errors, punctuation and typos.
We have corrected grammatical errors, punctuation, and typos.
(6) The first paragraph of the mathematical modeling section (methods) is jargon heavy and does not sufficiently convey a message to the general reader population.
Thanks for bringing this point to our attention. We have now extended the first paragraph of the mathematical modeling section in a way that it is more comprehensible to a general readership: “We propose a cell transition model describing the probability of naïve lymphocytes to transmigrate into the CSF or differentiate. The transition model can be rigorously derived using Markov jump processes describing differentiation and migration. A detailed derivation
has been shown in Ruck et al.. Therein, the model has been derived by first principles and applying techniques developed for large particle systems in gas dynamics. We propose a novel model in a similar spirit. A formal derivation can be done analogously to the procedure outlined in. We assume four possible stages, two for the compartments (PB, CSF) and two for naïve and differentiated lymphocytes. In each stage lymphocytes undergo a transition to another stage within a certain time interval with a certain probability. For example, a naïve lymphocyte from the PB migrates into the CSF while remaining naïve with the probability p = a1 (Fig. 1A, C). We model the transition of lymphocytes to different stages as a time independent process. Furthermore, we assume that the cells transition independently of each other, because there
are sufficient cells available. The transition model contains jump (i.e. transition) rates that are calibrated using experimental data as described below.“
(7) The sentence regarding CD19low cells (l. 203) does not make any sense and does not match the figure. Are cells possibly absent and not abundant?
Of course, the reviewer is right and the plasmacytoid cells are absent under homeostatic conditions. Accordingly, we have changed “abundant” into “absent”.
(8) Spell out abbreviations when a term is first used (RRMS).
We have doubled checked that all abbreviations have been spelled out at first use.
(9) Supplementary figure: To clarify gating strategy, add arrows to make succession more clear.
We agree with the reviewer that this is a good idea and have now added arrows to Supplemental Fig. 1.
(10) Line 260: Do the authors mean CD56 instead of CD52?
Here we are referring to alemtuzumab a monoclonal antibody directed towards the CAMPATH- 1 antigen (= CD52) a protein present on the surface of mature lymphocytes. We have now added the following sentence to clarify this point in the result section “While natalizumab prevents transmigration of lymphocyte subsets in the CNS by blocking the cell adhesion molecule alpha 4 integrin (Fig. 4A, left), alemtuzumab depletes CAMPATH-1 (CD52) expressing lymphocyte subsets in the periphery (Fig. 4A, right). Of note, CD52 is mainly expressed on mature lymphocytes.”
(11) Figure 4: It would be better to stick to a schematic alpha/ beta instead of introducing ABCD
As introduced in Fig. 1 a/b is the transition rate quantifying the change from one stage of a cell into another (e.g. transition from a naïve cell into a differentiated one), whereas A, B, C, and D describes the different stages of a cell with A = PB/naïve, B = CSF/naïve, C =PB/differentiated, and D = CSF/differentiated. However, since we also used the numbers 1, 2, 3, and 4 to describe the different stages we agree with the reviewer that the letters are redundant and deleted them. The text has been changed at the respective sections accordingly. Furthermore, we added the transition rates to Fig. 4
Reviewer 2 Report
The manuscript „Generation of a model to predict differentiation and migration of lymphocyte subsets under homeostatic and CNS autoinflammatory conditions“ by Gross et al suggests a mathematical model to predict cell distribution, migration, and differentiation between peripheral blood (PB) and cerebrospinal fluid (CSF) in steady state as well as in inflammatory diseases of the CNS.
The study builds up on previous work from the same group, where they proposed a system to model CD4 T cell distribution between blood and CSF during steady state (Ruck et al., Math Med Biol 2017). Here, they extend this model to additional immune cell subsets and activation stages, and include untreated, Natalizumab or Alemtuzumab treated MS patients as well as patients with Susac syndrome (SUS): in a first step, the authors calibrate the model system in non-inflammatory controls and extend it to relapsing-remitting Multiple Sclerosis (MS) patients. Here, the model correctly reflects PB and CSF findings, as well as transition states and migratory rates. In addition, treatment-induced changes by Natalizumab and Alemtuzumab, two paradigmatic treatment strategies for MS, are correctly modelled in the proposed system. Finally, they authors use SUS patients with both PB and CSF data to calibrate the model for this disease and, in the final step, to predict CSF changes based on the PB data.
General critique:
The study presented by Gross et al presents a novel mathematical system to model immune cell populations, their distribution, migration and differentiation, between PB and CSF in health and autoimmune diseases. The authors also include calculations as to treatment-induced alterations by specific immune-cell targeting therapies. Finally, they extend their findings to SUS, a rare autoimmune inflammatory CNS disease, and show the power and potential of this approach by predicting CSF alterations based on PB data.
This work is of high importance to the field, as it suggests to provide a novel tool to extrapolate CSF data from PB cell distributions. This is of great importance in MS, since CSF acquisition is done only rarely on routine basis during the course of the disease due to the invasiveness of the lumbar puncture. Yet, extrapolation of this data from PB would offer novel insight and might guide treatment decisions.
Thus, I consider this work of high novelty and great relevance to the field, both for basic researchers studying MS pathology as well as clinicians treating MS patients.
However, the major restriction to this work consists of the fact that the accuracy of the prediction of CSF data from PB data only is limited to SUS patients, while MS patients and controls are used to calibrate the model only. Thus, in order to increase the power of this novel model, I would appreciate if the authors included three examples of healthy controls and MS patients, respectively, where they have both PB and CSF data at hand, but calculate the CSF immune cell distribution based on the PB data only. Since this approach would reflect the “real world” situation, where CSF data will be predicted from PB, this dataset would greatly enhance the importance of this model system. If these examples are included and validate the model, I highly recommend publication of this important study.
As a minor point: there is a typo on p6, line 159: “naïve” should be exchanged to “differentiated”.
Author Response
Reviewer 2
The manuscript „Generation of a model to predict differentiation and migration of lymphocyte subsets under homeostatic and CNS autoinflammatory conditions“ by Gross et al suggests a mathematical model to predict cell distribution, migration, and differentiation between peripheral blood (PB) and cerebrospinal fluid (CSF) in steady state as well as in inflammatory diseases of the CNS.
The study builds up on previous work from the same group, where they proposed a system to model CD4 T cell distribution between blood and CSF during steady state (Ruck et al., Math Med Biol 2017). Here, they extend this model to additional immune cell subsets and activation stages, and include untreated, Natalizumab or Alemtuzumab treated MS patients as well as
patients with Susac syndrome (SUS): in a first step, the authors calibrate the model system in non-inflammatory controls and extend it to relapsing-remitting Multiple Sclerosis (MS) patients. Here, the model correctly reflects PB and CSF findings, as well as transition states and migratory rates. In addition, treatment-induced changes by Natalizumab and Alemtuzumab, two paradigmatic treatment strategies for MS, are correctly modelled in the proposed system. Finally, they authors use SUS patients with both PB and CSF data to calibrate the model for
this disease and, in the final step, to predict CSF changes based on the PB data.
General critique:
The study presented by Gross et al presents a novel mathematical system to model immune cell populations, their distribution, migration and differentiation, between PB and CSF in health and autoimmune diseases. The authors also include calculations as to treatment-induced alterations by specific immune-cell targeting therapies. Finally, they extend their findings to SUS, a rare autoimmune inflammatory CNS disease, and show the power and potential of this approach by predicting CSF alterations based on PB data.
This work is of high importance to the field, as it suggests to provide a novel tool to extrapolate CSF data from PB cell distributions. This is of great importance in MS, since CSF acquisition is done only rarely on routine basis during the course of the disease due to the invasiveness of the lumbar puncture. Yet, extrapolation of this data from PB would offer novel insight and
might guide treatment decisions.
Thus, I consider this work of high novelty and great relevance to the field, both for basic researchers studying MS pathology as well as clinicians treating MS patients. However, the major restriction to this work consists of the fact that the accuracy of the prediction of CSF data from PB data only is limited to SUS patients, while MS patients and controls are used to calibrate the model only. Thus, in order to increase the power of this novel model, I would appreciate if the authors included three examples of healthy controls and MS patients, respectively, where they have both PB and CSF data at hand, but calculate the CSF immune cell distribution based on the PB data only. Since this approach would reflect the “real world” situation, where CSF data will be predicted from PB, this dataset would greatly enhance the importance of this model system. If these examples are included and validate the model, I highly recommend publication of this important study.
We agree with the reviewer that it is a good idea to perform this important analysis. Since we have used all our non-inflammatory controls calibrating the model, we unfortunately cannot perform this analysis with healthy controls. However, we were able to identify 5 therapy naïve MS patients who were not included in our calibration cohort and calculated the CSF data based
on the PB data only as suggested by the reviewer (new Fig. 3C). Since the model was calibrated using the whole patient cohorts, the predicted data vary from the actual ones as expected. Nevertheless, overall the model predicted the distribution of lymphocyte subsets within the CSF compartment. We elaborated this limitation of the model in the discussion:
“However, although the model is precise to predict disease-specific distributions predicted data may vary from the acquired ones for individual patients, because the model was calibrated using the whole patient cohort.”
As a minor point: there is a typo on p6, line 159: “naïve” should be exchanged to “differentiated”.
Thanks for bringing this to our attention. We corrected the typo accordingly.
Round 2
Reviewer 2 Report
In the revised version of their manuscript, the authors now display cell numbers of 5 additional therapy naive patients, who were not included in the initial calibration cohort, to predict CSF changes based on the peripheral blood cell distribution.
In this "validation cohort", the model predicts numbers of differentiated CD4, B, and NK cells sufficiently. However, the prediction of individual subpopulations varies especially for naive CD4 as well as naive and differentiated CD8 T cells. The authors now pointed out to acknowledge this restriction in the revised version of the manuscript.
However, even though the new Figure 3C is shown, it is not mentioned in the text. Also, the respective Figure legend is missing. Moreover, I cannot find the mentioned explanation in the Discussion.
If these parts are added to a final revised version of the manuscript, I find this solution satisfying and recommend publication of the manuscript.
Author Response
We would like to thank the Reviewer to bring this point to our attention. It seems like the revised document file provided by the Assistant Editor contained our new figure, but none of the other changes we have applied to the text. We apologize for the inconvenience this issue has caused.
To play it save this time we also added a PDF File highlighting our changes as an attachment below.
Within that document The mentioned changes are in the result section (page 5-6, line 172-176), Figure legend 3 (page 6, line 187-190), and discussion (page 10, line 289-291).
